

# Conditional generative models for sampling and phase transition indication in spin systems

Japneet Singh [1], Mathias S. Scheurer [2,3*] and Vipul Arora [1]

**1** Department Of Electrical Engineering, Indian Institute Of Technology Kanpur, Kanpur, Uttar Pradesh, India
**2** Institute for Theoretical Physics, University of Innsbruck, Innsbruck A-6020, Austria
**3** Department of Physics, Harvard University, Cambridge MA 02138, USA

⋆ Mathias.Scheurer@uibk.ac.at

## Abstract

In this work, we study generative adversarial networks (GANs) as a tool to learn the distribution of spin configurations and to generate samples, conditioned on external tuning parameters or other quantities associated with individual configurations. For concreteness, we focus on two examples of conditional variables—the temperature of the system and the energy of the samples. We show that temperature-conditioned models can not only be used to generate samples across thermal phase transitions, but also be employed as unsupervised indicators of transitions. To this end, we introduce a GAN-fidelity measure that captures the model's susceptibility to external changes of parameters. The proposed energy-conditioned models are integrated with Monte Carlo simulations to perform over-relaxation steps, which break the Markov chain and reduce auto-correlations. We propose ways of efficiently representing the physical states in our network architectures, e.g., by exploiting symmetries, and to minimize the correlations between generated samples. A detailed evaluation, using the two-dimensional XY model as an example, shows that these incorporations bring in considerable improvements over standard machine-learning approaches. We further study the performance of our architectures when no training data is provided near the critical region.



# 1 Introduction

Generative models [1–4] aim at modelling complicated probability distributions of data in a way that they can readily be used to generate new samples. These techniques model the joint distribution of data, such as images of handwritten digits, and some useful quantities associated with the data, e.g., which of the ten digits is shown. The model is then used to generate unseen data by sampling from the learnt joint probability distribution, e.g., produce unseen images of digits.

In physics, we often start from a Hamiltonian, an action, or just a classical configuration energy, describing the system of interest, and, as such, formally, know the distribution of the elementary degrees of freedom, such as the fields in a field theory or the spin configurations in a classical spin model. Typically, one is interested in studying the behavior of these distributions as a function of tuning parameters, e.g., temperature or coupling constants, and one can

think of them as the distribution of data conditioned on these tuning parameters. Since, however, this data is usually very high-dimensional, the essential physical properties can only be captured by evaluating physical quantities, such as symmetry-breaking order parameters and their susceptibilities, or non-local probes of topological properties. In most interesting cases, their evaluation cannot be performed analytically and, hence, numerical techniques have to be used. Among those, in particular, Monte Carlo methods, where observables are estimated by sampling from the data, are powerful, as they, at least in principle, guarantee asymptotic convergence to the true distribution.

Markov chain Monte Carlo (MCMC) techniques work by constructing a first order Markov sequence where the next sample is dependent on the current sample. Unfortunately, these methods can suffer from the problem of large thermalization times and large auto-correlation times (especially near phase transitions), both of which increase drastically with the increase in lattice size. For quickly generating uncorrelated samples, we need the auto-correlation time to be small. Starting from a random configuration, for efficiently reaching the state of generating valid samples that conform to the underlying true distribution, the thermalization time has to be short as well.

To curtail the effect of dramatic increase of auto-correlation time near criticality, many global update methods have been developed, which simultaneously change the variables at many sites in a single MC update, such as Swendsen-Wang [5], Wolff [6], worm [7], loop [8,9] and directed loop [10,11] algorithms. But these methods work only for specific types of models and not for any generic system.

Besides several other promising applications of machine-learning methods in physics [12–16], generative modelling techniques have been explored for enhanced generalizability and performance. For instance, Efthymiou and Melko [17] use deep-learning-based super-resolution techniques to produce spin configurations of larger system sizes from MCMC-generated configurations of smaller sizes by the use of convolutional neural networks (CNNs). The resolved configurations have thermodynamic observables that agree with Monte-Carlo calculations for one and two-dimensional (2D) Ising models. Another approach is 'self-learning Monte Carlo' [18–21] that, in principle, works for any generic system and applies machine-learning-based approaches on top of MCMC to speed up the simulations and to reduce the increase in auto-correlation time near the critical temperature. Other approaches which apply machine-learning techniques as a supplement or alternative to MCMC are based on normalizing flow [22], Boltzmann machines [23–26], on reinforcement learning [27], on generative adversarial networks (GANs) [28–33], autoencoders [34–36], and on variational autoregressive networks [37–40].

So far, in most of these approaches, the underlying generative model is trained separately for different values of the tuning parameters of the system, such as different temperatures. But when configurations for multiple temperatures, including close to criticality, need to be generated, either they require configurations for that corresponding temperature and training a model again and/or the Markov chain has to be re-started altogether. For this reason, we here explore a different and less used [31–33] strategy, which consists of learning the *conditional* probability distribution of physical samples, conditioned on a (in general set of) parameter(s) $c$.

One can distinguish two different types of conditional parameters relevant for physical models: $c$ can either be an external tuning parameter, such as temperature for a thermal phase transition or coupling constants in a model, or a quantity that is associated with and a unique function of each sample, such as its energy or the number of topological defects in it. In this work, we study an example of each of the two types of $c$: temperature-conditioned and energy-conditioned models. In the former case, as the name suggests, we provide temperature as conditional information in the training data set (obtained via MCMC) for our

deep-learning-based conditional generative models. Most notably, these include conditional GANs [41], among other models employed as baselines. After training, our models are used to generate samples at different temperatures, which are not necessarily equal to the values of temperature in the training data set. For our energy-conditioned models, we show how they can be integrated with MCMC and can be used for additional over-relaxation steps which break the Markov chain and dampen auto-correlations. They are well-suited for this purpose, as they can quickly sample configurations with energy close to the energy of the current sample in the Markov chain while being locally dissimilar. We also study the performance of these two different applications when the training data is limited to temperatures away from the transition. Due to the generality of our approach, we believe that the optimization strategies for generative modeling of physical systems we discuss in this work will also be useful for the application to experimentally generated data [33, 42].

Generative models can be broadly subsumed into two categories—prescribed and implicit [43]. *Prescribed models* are those that provide an explicit parametric specification of the distribution of the output (data). These models typically deploy Bernoulli or Gaussian outputs, depending on the type of data. On the other hand, *implicit models* directly generate data by passing a noise vector through a deterministic function which is generally a neural network. Implicit models can be more expressive than their prescribed counterparts but calculating likelihood becomes intractable in most cases. Most of the generative models in machine learning are prescribed models as they have a notion of likelihood, are easy to optimize and produce excellent results. But, generally, they make an assumption of independence between the parametric distribution across various pixels or lattice sites. Such assumptions in physics can be quite restrictive as the models need to capture the correlations between lattice sites. Prescribed models would otherwise need to estimate large co-variance matrices and ensure their positive-definiteness. For this reason, we expect and also confirm by our numerical experiments that implicit generative models, in particular in the GAN framework, are more suitable for modelling the site-to-site correlations in physical systems.

Additionally, we propose other modifications that exploit the underlying structure of the physical systems and enhance the model's utility. The proposed modifications can bring significant improvement in performance as compared to the prescribed models treated as baselines. We also show that, for implicit models, maximizing the mutual information between a set of structured latent variables and reconstructed configurations leads to maximizing a lower bound on the entropy of the learnt distribution; this reduces the correlations among configurations generated by the model and can act as an indicator of phase transitions. We evaluate in detail the improvements in performance of the various modifications we propose. While our approaches can be readily applied to other systems as well, we focus for concreteness in our numerical studies on the 2D XY model, as it provides a transparent example to benchmark these modifications and has been established as a challenging model for neural networks [44].

If the type of phase transition and the associated observable, e.g., a local order parameter, are known, these quantities can be evaluated with the generated samples to capture the phase transition. For instance, in case of the XY model, the finite-temperature BKT transition is associated with the proliferation/suppression of vortices [45–48]. While we show that our generative models can indeed reproduce the expected behavior of vortices, we also demonstrate that our trained network can be used to reveal the transition without requiring knowledge about the underlying nature of the phase transition. This unsupervised detection of phase transitions is another central topic of machine learning in physics. In particular, topological transitions, such as the BKT transition, are challenging due to their non-local nature; however, the method proposed in [49] has been demonstrated to work in a variety of different models [49–51] and extensions [52] for symmetry-protected topological phases have been developed. We here demonstrate that trained generative models can also be used to indicate the phase transition

in an unsupervised way: as expected [53–56], we find that the model is particularly suscep­tible to parameter changes in the vicinity of the transition. We quantify this by introducing a fidelity measure constructed on the trained GAN that can be efficiently evaluated and shows peaks in the vicinity of the phase transition.

The remainder of this paper is organized as follows. In Sec. 2, we provide an introduction to the different generative modelling techniques we explore in this work and to the XY model. The modifications we propose for an effective modelling of physical systems are described in detail in Sec. 3. The numerical experiments, using the XY model as concrete example, are presented in Sec. 4. Finally, Sec. 5 contains a brief summary.

# 2 Generative modelling and XY model

To establish notation and nomenclature, we first provide an introduction to the generative machine-learning methods we use—variational autoencoders (VAEs) and GANs, as well as their conditional extensions; we also define the 2D XY model, which is the model we use to benchmark our machine learning approach with, and the physical quantities we study. Readers familiar with the XY model and these generative machine-learning techniques, can skip this section and proceed directly with Sec. 3.

## 2.1 Variational autoencoders

VAEs are powerful continuous latent variable models used for generative modelling of a high-dimensional distribution over a given data set, allowing one to sample directly from the data distribution [57]. They have shown promising results in producing unseen fake images and audio files which are almost indistinguishable from real data, see Ref. [58] for instance. In its standard form, a VAE consists of an encoder and a decoder. The encoder maps from data space $X$ to a latent space $z \subseteq \mathbb{R}^D$ and consists of a family of distributions $\mathbb{Q}_\phi$ on $z$ parameterized by $\phi$; it is typically modeled by deep neural networks. The decoder consists of a family of distributions $\mathbb{P}_\theta$ on $X$ parameterized by $\theta$. As the name implies, the encoder encodes the semantic information present in the data into the latent space. The decoder uses the encoded information in latent space to reconstruct the data. The overall objective is to maximize the likelihood of the data, independently and identically distributed as $P(x) = \int P_\theta(x|z)P(z)dz$, where, $x \in X$, $z \in z$, $P_\theta(x|z) \in \mathbb{P}_\theta$, and $P(z)$ is the prior distribution, often taken as Gaussian. The likelihood is generally intractable to compute but can be maximized by maximizing the evidence lower bound (ELBO). The ELBO for marginal log-likelihood $P_\theta(x)$ for a data-point $x$ is expressed as

$$\log P_\theta(x) \geq \mathbb{E}_{z \sim Q_\phi(z|x)}[\log P_\theta(x|z)] - D_{\text{KL}}[Q_\phi(z|x)||P(z)],$$

where $Q_\phi(z|x) \in \mathbb{Q}_\phi$. The ELBO consists of 2 terms: $(i)$ a loss term accounting for the error in the reconstructed data and $(ii)$ a regularizing term which makes the encoder to encode information such that its distribution is close in Kullback-Leibler (KL) divergence, $D_{\text{KL}}$, to the prior distribution $P(z)$.

**Conditional VAE (C-VAE)** is a simple extension of standard VAE, with the only difference that the data distribution as well as the latent distribution are both conditioned by some ex­ternal information. We illustrate the typical structure of a C-VAE in Fig. 1a. The objective is now to maximize the likelihood conditioned on a given conditional information $c$. For our purposes here of generating samples of a physical model, the "conditional information" refers to the tuning parameters of interest in that model, such as temperature, $T$, ratios of exchange interactions in spin models, and the energy of samples, which can be used for sampling of the

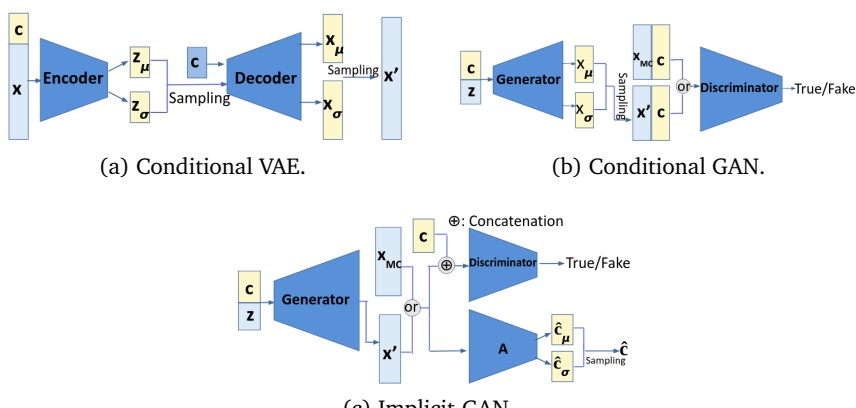

(a) Conditional VAE.  (b) Conditional GAN.

(c) Implicit-GAN.

Figure 1: Block-diagram representation of (a) C-VAE, (b) C-GAN, and (c), our proposed method, an Implicit-GAN. We refer to the respective parts of the main text, Sec. 2.1, Sec. 2.2, and Sec. 3.2, for a detailed description.

corresponding microcanonical ensemble or, as we will demonstrate below, decorrelate regular MCMC schemes by providing efficient overrelaxation steps. In general, $c$ can be a multi-component vector comprising several physical tuning parameters or quantities associated with the individual samples.

To train the C-VAE, we again maximize the ELBO, now assuming the form

$$\log P_\theta(x|c) \geq \mathbb{E}_{z \sim Q_\phi(z|x,c)}[\log P_\theta(x|z,c)] - D_{\mathrm{KL}}[Q_\phi(z|x,c)||P(z|c)].$$

Here, we will assume the prior distribution to be independent of $c$ and to follow a normal distribution with zero mean and variance 1, i.e., $P(z|c) = P(z) = \mathcal{N}(0,I)$.

## 2.2 Generative adversarial networks

GANs [59] are another powerful framework for modelling a probability distribution. In physics, GANs have been successfully applied to many different models ranging from binary spin systems like the Ising model [29], to the Fermi-Hubbard model [33], high-energy physics [28], cosmology [60], and material science [30]. A GAN consists of two models, a generator $G(z)$ and a discriminator $D(x)$. The generator is a function $G : z \to X$ which tries to capture the data distribution and produces samples $x$ that closely resemble samples from the training data. On the other hand, the discriminator is a function $D : X \to (0,1)$ which tries to estimate the probability that a sample came from the true data distribution (true sample) rather than from the generative model $G$ (fake/negative sample). $G$ tries to maximize the probability of $D$ making a mistake while $D$ tries to minimize the probability of being fooled by $G$. The result is a minimax game between two players, described by the value function

$$V(G,D) = \mathbb{E}_{x \sim p_{\mathrm{Data}}}[\log D(x)] + \mathbb{E}_{z \sim p(z)}[\log(1 - D(G(z)))]. \tag{1}$$

The objective of this game can be expressed as $\min_G \max_D V(G,D)$.

**Conditional GANs (C-GANs)** are a simple extension [41] of standard GANs in which the generator produces samples based on the external information $c$ while the discriminator tries to estimate the probability that the sample came from the true conditional data distribution rather than from $G$. The associated minimax objective now becomes

$$\min_G \max_D V(G,D;c) = \min_G \max_D \left( \mathbb{E}_{x \sim p_{\mathrm{Data}}}[\log D(x;c)] + \mathbb{E}_{z \sim p(z)}[\log(1 - D(G(z;c);c))] \right) \tag{2}$$

and we show the basic structure of a C-GAN in Fig. 1b.

## 2.3 2D XY model

While the methods we propose and compare in this work are more generally applicable, we will employ one specific physical model, the classical 2D XY-spin model, to illustrate and test the generative machine-learning methods. The XY model was chosen as it features key challenges—compact local degrees of freedom (two-component units vectors) and non-local, topological excitations (vortices) together with conventional excitations (spin waves)—in a minimal setting. At the same time, it is accessible via conventional MCMC sampling schemes, which is important for us since it allows to test the accuracy of our generative models.

More specifically, the XY model consists of two-component spins on every site $i$ of the lattice with fixed magnitude, which we set to 1 and, hence, are described by the unit vectors $s_i = (\cos \theta_i, \sin \theta_i)^T$, $\theta_i \in [0, 2\pi)$. We here consider a 2D square-lattice of size $N \times N$ and restrict ourselves to ferromagnetic nearest-neighbor interactions, $J > 0$; using the latter as unit of energy, $J \equiv 1$, the energy of a configuration $\boldsymbol{\theta} = \{\theta_i\}$ is given by

$$E(\boldsymbol{\theta}) = -\sum_{\langle i,j \rangle} s_i \cdot s_j = -\sum_{\langle i,j \rangle} \cos(\theta_i - \theta_j), \tag{3}$$

where the sum over $\langle i, j \rangle$ includes all the adjacent sites on the lattice.

The probability density of a configuration $\boldsymbol{\theta}$ at a given temperature $T \in \mathbb{R}^+$ is given by

$$P_T(\boldsymbol{\theta}) = \frac{1}{Z(T)} e^{-\frac{E(\boldsymbol{\theta})}{T}}, \tag{4}$$

where the Boltzmann constant is set to unity and $Z(T) = \sum_{\boldsymbol{\theta}} e^{-\frac{E(\boldsymbol{\theta})}{T}}$ is the partition function. Thermal expectation values, $\langle \mathcal{O} \rangle_T$, of physical quantities $\mathcal{O} = \mathcal{O}(\boldsymbol{\theta})$, such as mean magnetization, $\boldsymbol{m}(\boldsymbol{\theta}) = N^{-2} \sum_i s_i(\theta_i)$, or mean energy, $e(\boldsymbol{\theta}) = N^{-2} E(\boldsymbol{\theta})$, follow from Eq. (4) as

$$\langle \mathcal{O} \rangle_T = \sum_{\boldsymbol{\theta}} \mathcal{O}(\boldsymbol{\theta}) P_T(\boldsymbol{\theta}). \tag{5}$$

In general, Eq. (5) cannot be evaluated exactly and, hence, has to be analyzed with approximate analytical techniques or numerical approaches. One of the most common ways of evaluating the sum in Eq. (5) numerically, proceed via MCMC sampling of configurations $\boldsymbol{\theta}$ according to the distribution $P_T(\boldsymbol{\theta})$, e.g., via the Metropolis-Hastings (MH) algorithm [61]. In each step of the MH algorithm, a configuration $\boldsymbol{\theta}'$ is generated from a current configuration $\boldsymbol{\theta}$ with some *a priori* selection probability $W(\boldsymbol{\theta}'|\boldsymbol{\theta})$. This new configuration is then accepted with probability

$$W_A(\boldsymbol{\theta}'|\boldsymbol{\theta}) = \min\left(1, \frac{W(\boldsymbol{\theta}|\boldsymbol{\theta}')e^{-E(\boldsymbol{\theta}')}}{W(\boldsymbol{\theta}'|\boldsymbol{\theta})e^{-E(\boldsymbol{\theta})}}\right). \tag{6}$$

When $W(\cdot)$ is symmetric, i.e., $W(\boldsymbol{\theta}|\boldsymbol{\theta}') = W(\boldsymbol{\theta}'|\boldsymbol{\theta})$, then Eq. (6) becomes

$$W_A(\boldsymbol{\theta}'|\boldsymbol{\theta}) = \min(1, e^{-(E(\boldsymbol{\theta}')-E(\boldsymbol{\theta}))}). \tag{7}$$

The goal of this work is to investigate how generative models can be used to generate samples $\boldsymbol{\theta}$ for efficient evaluation of the expectation values of observables in Eq. (5).

Besides the mean energy and magnetization mentioned above, we also investigate the number of vortices in the system at a given temperature. Vortices are non-local excitations defined by a non-zero winding, $v \neq 0$, of the unit vector $s_i$ on any closed path encircling

the core of the vortex. Proliferation or suppression of vortices are the defining feature for the finite-temperature phase transition, the BKT transition [45–48], of the 2D XY model. Studying vortices is not only motivated by the fact that they are integral to the physics of the XY model, but also due to their non-local, topological nature; as a consequence, one might expect that vortices are more difficult to capture by machine-learning techniques than local excitations.

In practice, we detect vortices in samples by counting, for every site $i$, the angle differences in anti-clockwise sense around the $(3 \times 3)$ square centered at $i$. Each difference was constrained to lie in $(-\pi, \pi]$ using a saw function. The "vorticity" $V$ of a configuration $\boldsymbol{\theta}$ is the number of vortices with winding number $v = +1$.

## 3 Proposed method

Having introduced the basic generative models, we will next discuss our proposed implementation and some additional modifications which improve the models' performance in generating samples. To be concrete, we will discuss them mostly in the context of the 2D XY model, although they apply equally well to many other systems as well. These modifications are motivated from the structure of the physical system. First, we discuss how the states are represented in both of our implementations. Then we detail the changes in the models' structures and training objectives. To analyze systematically how relevant the different modifications are, we present an ablation analysis in Appendix A.

### 3.1 Representation of physical states

The first set of modifications concerns the representation of states. As we will see, choosing a proper way of parameterizing the physical states is integral to an efficient and feasible generative modelling.

#### 3.1.1 Exploiting symmetries

First of all, many physical systems exhibit symmetries. Formally, this means that the energy $E(x)$ of any state $x$ is the same as that of the transformed state, $x'$, $E(x) = E(x')$. This can be exploited to find a more compact representation of the state: one can represent states such that two states that are related by a symmetry have the exact same representation. Unbiased sampling is guaranteed by randomly performing symmetry transformations on the generated state, since $E(x) = E(x')$ implies that any two symmetry-related states are equally likely.

In the case of the XY model, an important symmetry is the invariance under global rotation of all spins,

$$\boldsymbol{\theta} \quad \longrightarrow \quad \boldsymbol{\theta}' = \boldsymbol{\theta} + \theta_0, \quad \theta_0 \in \mathbb{R}. \tag{8a}$$

This symmetry allows us to reduce the dimensionality of the representation of the states from $N^2$ to $N^2 - 1$. In practice, for any given state $\boldsymbol{\theta}$ we choose $\theta_0$ such that

$$\left(\boldsymbol{m}(\theta_i')\right)_y = N^{-2} \sum_i \sin(\theta_i') = 0, \tag{8b}$$

i.e., describe the state by deviations of the spin orientations about a certain 'mean-direction' (here chosen along the $x$-axis). As $E(\boldsymbol{\theta})$ for the XY model is invariant under Eq. (8a), we know $P_T(\boldsymbol{\theta}) = P_T(\boldsymbol{\theta}', \theta_0) = P(\boldsymbol{\theta}')P(\theta_0)$, with uniform $P(\theta_0)$. We will model $P(\boldsymbol{\theta}')$ using a deep generative model, and sample $\theta_0$ uniformly in $[0, 2\pi)$. Thus, we have reduced the dimensionality of space (the degrees of freedom of data) in which the manifold of lattice configurations is embedded and, more importantly, made sure that the symmetries are respected *exactly* by our sampling procedure.

### 3.1.2 Topology of degrees of freedom

For many physical systems, the degrees of freedom on every site are compact. For instance, for XY-spin or Heisenberg-spin models, the local configuration space is a one-dimensional or two-dimensional sphere, respectively. In these cases, one has to be careful about choosing a smooth representation of these spaces that respects their topology.

For the XY, the angles $\theta_i \in [0, 2\pi)$ have discontinuous jumps at $2\pi$. As such, directly using angles as input to the model does not explicitly take into account the topological and geometrical properties of the space of XY spins. For example, an angle of $2°$ is quite similar to $358°$, and also $180°$ is not a good estimate of the mean spin orientation. The topology at each lattice site can be taken into account by using a two-channel lattice consisting of cosines and sines of lattice angles at both input and output of our model; this means that instead of $\theta_i$, we use the two-component unit vectors $\boldsymbol{s}_i = (\cos\theta_i, \sin\theta_i)^T$, as has previously been implemented for different machine learning studies of the XY model (see, e.g., Ref. [62]).

Such a choice of input and output makes the model an implicit model. This also allows to overcome the limitations on the model's ability to capture correlations between lattice sites due to independent sampling from $N(\mu_i, \sigma_i)$ at each lattice site $i$. We use this representation for the GAN framework. A similar extension to VAE framework makes the ELBO intractable. While there exist approaches like that of Ref. [63] to overcome this issue, most of them are based on adversarial training (or likelihood free inference).

### 3.1.3 Periodic boundary conditions

As we are interested in the bulk properties of the XY model and not in the behavior around edges, we will assume periodic boundary conditions throughout this work. Mathematically, this means that we replace $\theta_{(i_1,i_2)}$, by $\tilde{\theta}_{(i_1,i_2)} := \theta_{((i_1)_N, (i_2)_N)}$, where $(i)_N$ denotes $i$ modulo $N$. For the implementation with deep neural networks, we increase the size of the lattice from $N \times N$ to $(N+2) \times (N+2)$, keeping the middle $N \times N$ lattice sites the same and filling the sites at the new edges in accordance with the periodic boundary conditions. We expect that this improves the performance of feature extracting kernels of the CNNs especially at the "edges" of a lattice. We use this form of periodic padding on the input layer of the encoder (for VAE) or discriminator (for GAN).

## 3.2 Proposed conditional models

Now, we describe the proposed implicit GAN models for lattice simulations. The ImplicitGAN can be conditioned on temperature or on energy, which we denote as "ImplicitGAN-$T$" and "ImplicitGAN-$E$", respectively.

### 3.2.1 Minimizing output biases

As mentioned above, we propose to normalize the spin configurations such that their net magnetization vector $\boldsymbol{m}(\boldsymbol{\theta}')$ always points along the x-axis, see Eq. (8). But, there is nothing in the training objective in Eq. (2) which explicitly incentivizes the network to produce configurations with their magnetization to point along the x-axis. If this condition is not satisfied, it implies that our model has developed some bias, which may be due to the model parameters being stuck in a local minimum during training. We indeed observed that the training objective in Eq. (2) can lead to bad local optima, as discussed later in Sec. A. Thus, if we add a term forcing the generative model to minimize the square of the $y$-component of the magnetization

in a configuration we can minimize such biases. The GAN value function now becomes

$$V_b(G,D;c) = V(G,D;c) + \lambda \, \mathbb{E}_{z \sim p(z); \boldsymbol{\theta}' = G(z;c)} \left[ \sum_i \frac{\sin(\theta_i')}{N^2} \right]^2, \tag{9}$$

where $\lambda \in \mathbb{R}^+$ is a constant hyper-parameter.

### 3.2.2 Maximizing the output entropy

The generated samples will hardly have any practical significance if we cannot guarantee convergence to the exact distribution—especially considering the fact that GANs are susceptible to the mode-collapse problem, i.e., they might miss a subset of the modes of a multimodal distribution of the samples. In practice, we could use the generated $x$ as the initial configuration for MCMC. But if the different samples generated by our model have high correlations among themselves, the number of MCMC steps needed to obtain uncorrelated samples would be large, thereby, defeating the purpose of the extra computational efforts for training the generator. We can decrease the number of MCMC steps needed if we can reduce the initial correlation among the different samples generated by our model.

To achieve this, we propose to additionally maximize the overall entropy (more specifically, the 'differential entropy') of the learnt distribution $h(G(z,c))$, i.e., to make the learnt distribution more 'diffused', while also keeping the distribution of generated samples in close agreement to the true distribution for all temperatures. It has been shown that, in the case of prescribed models, the entropy-regularized loss function reduces the problem of mode-collapse [64]. In practice, the problem is that $h(x)$ is difficult to compute or maximize. However, we can instead maximize a lower bound on $h(x)$ in the following way: due to the symmetry, $I(x;c) = I(c;x)$, of the mutual information $I$, it holds

$$h(x) = h(c) - h(c|x) + h(x|c)$$
$$\geq h(c) - h(c|x) + h(x|c,z).$$

Now, $h(x|c,z) = 0$ for an implicit model (as opposed to prescribed models, where $h(x|c,z)$ may not be non-negative), because the value of $x$ is completely determined by the value of $\{c,z\}$. Thus,

$$h(x) \geq h(c) - h(c|x) = I(c;x). \tag{10}$$

Here $h(c)$ is constant because we have already specified and fixed the latent distribution of conditional information: in the case of temperature-conditioned models, $c \equiv T$ and $P(T)$ is uniform over all temperatures in the training data. For energy-conditioned models, we have $c \equiv E$ with $P(E)$ being determined by the physical system and the choice of training data. Consequently, minimizing $h(c|x)$ maximizes the lower bound on $h(x)$.

Minimizing $h(c|x)$ requires access to the posterior $P(c|x)$. But, we can minimize an upper bound on $h(c|x)$ by defining an auxiliary distribution $A(\hat{c}|x)$ as:

$$h(c|x) = -\mathbb{E}_x[\mathbb{E}_{c \sim P(c|x)}[\log P(c|x)]]$$
$$= -\mathbb{E}_x[D_{\mathrm{KL}}(P(\hat{c}|x) \| A(\hat{c}|x)) + \mathbb{E}_{\hat{c} \sim P(c|x)}[\log A(\hat{c}|x)]]$$
$$\leq -\mathbb{E}_x[\mathbb{E}_{\hat{c} \sim P(c|x)}[\log A(\hat{c}|x)]]$$
$$= -\mathbb{E}_{\hat{c} \sim P(c)}[\mathbb{E}_{x \sim P(x|c)}[\log A(\hat{c}|x)]]$$
$$\equiv L_H(G,A). \tag{11}$$

We use an auxiliary network $A$ to estimate $c$ from $x$, i.e., maximize the probability $P(\hat{c} = c)$. Such a technique of maximizing a lower bound on mutual information in terms of an auxiliary distribution was previously proposed in [65]. According to Eq. (11), $h(\hat{c}|x)$ can be minimized by minimizing its upper bound given by $L_H(G, A)$. Note the bound becomes tight when $\mathbb{E}_x[D_{\mathrm{KL}}(P(\hat{c}|x)||A(\hat{c}|x))] \to 0$. The modified objective, which involves the auxiliary distribution, is given by

$$\min_{G,A} \max_{D} \{V_b(G, D; c) + \gamma L_H(G, A)\}, \tag{12}$$

where $\gamma \in \mathbb{R}^+$ is a constant hyper-parameter and $V_b$ as in Eq. (9). Note $L_H(G, A)$ maximizes only a lower bound on the entropy and, hence, $h(x)$ is not guaranteed to increase. The gap $h(x|c) - h(x|c, z) = I(x; z|c)$ is expected to be small since, by the structure of the model, one does not expect large mutual information between noise variables and generated samples. Since $I(x; z|c) \geq 0$, the overall entropy is likely to increase in practice.

Typically, $A$ and $D$ are implemented as neural networks sharing most of the layers. But, in our case, the information of $c$ should only be given to $D$ and not to $A$. Therefore, they were employed as separate neural networks, as shown in Fig. 1c. The discriminator $D$ tries to predict the probability that the sample belongs to the true distribution, while the auxiliary network $A$ outputs a distribution over $c$ for a given configuration. The distribution is assumed to be Gaussian with mean and variance $\hat{c}_\mu$ and $\hat{c}_\sigma$ predicted by the network $A$.

## 3.3 Unsupervised detection of phase transitions

So far, our focus has been on generating samples following Eq. (4) for the evaluation of physical observables according to Eq. (5). If we are interested in studying phase transitions and know which observables capture the transition, e.g., a local order parameter in case of a conventional, symmetry-breaking phase transition, we can simply evaluate these observables with our generated samples. However, one of the central questions of machine learning in the context of condensed matter and statistical physics is to find ways of detecting the transition without "telling" the algorithm which observables are relevant. The in this sense "unsupervised" detection of phase transitions could potentially be useful in cases where the order parameter or topological invariant characterizing the transition are not known.

Having constructed models that can generate samples at a given value of the conditional parameter(s) $c$, we here analyze whether the behavior of these models upon tuning $c$ can be used to infer where phase transitions take place, without requiring any knowledge about the underlying order parameter. In line with previous works [53–56], dealing with different machine-learning setups, we expect that our generative models are particularly susceptible to changes in $c$ in the vicinity of phase transitions. For ease of reading and since we explicit study this choice in our numerical experiments, we will use $c = T$ in the remainder of this subsection. We reiterate, however, that our machine-learning framework is able to provide samples subject to, in principle, arbitrary conditional constraints $c$. For instance, $c = E$ will allow studying transitions as a function of energy in a microcanonical ensemble or studying the behavior of the system as a function of other "post-selection" conditions on the samples is achievable as well.

The first measure we use is directly related to the one defined in previous works [53, 56] and makes use of the auxiliary network $A(x) = \hat{T}$ that we implemented to estimate the temperature from the samples $x$, needed to maximize the output entropy. One expects that the expectation value $\mathbb{E}_{x \sim P_T}[A(x)]$ over samples $x$ at temperature $T$ is approximately constant deep inside the two phases and that it varies maximally at the transition. As such

$$\mathcal{D}(T_0) = \frac{\partial \mathbb{E}_{x \sim P_T}[A(x)]}{\partial T}\bigg|_{T=T_0} \approx \frac{\mathbb{E}_{x \sim P_{T_0 + \Delta T}}[A(x)] - \mathbb{E}_{x \sim P_{T_0 - \Delta T}}[A(x)]}{2\Delta T} \tag{13}$$

should be peaked around the critical temperature.

The second measure we introduce is unique to GANs and can be defined for any GAN architecture, not only for the modified version with the additional auxiliary network. This measure is analogous to the widely studied quantum fidelity, which has also been extended to finite temperature and thermal phase transitions [66]. It is based on the idea that the form of a state (density matrix for thermal ensembles) will change most dramatically upon modifying a tuning parameter by a small amount (such as temperature $T \rightarrow T + \Delta T$) in the vicinity of a phase transition. This will first require a measure of similarity of two states or ensembles. For this we will use the expectation value of $D(x, T)$ with $x$ taken from some given ensemble $p'$. Since $D(x, T)$ estimates the probability of $x$ coming from the true thermal ensemble, this expectation value quantifies how similar the thermal ensemble and $p'$ are. Since we are interested in tuning temperature, we replace $p'$ by the ensemble generated by the generator at a different temperature and, thus, define the *GAN fidelity* as

$$\mathcal{F}_{\text{GAN}}(T) = \frac{1}{\Delta T} \mathbb{E}_{z \sim p(z)}[D(G(z; T), T) - D(G(z; T), T + \Delta T)]. \tag{14}$$

Imagine starting in the high-temperature phase and gradually decreasing $T$. Once $T$ reaches the phase transition, the generator in the second term in Eq. (14) starts producing samples that are not "expected" by the discriminator. Thus, the latter decreases its value, $\mathcal{F}_{\text{GAN}}(T)$ increases, and is expected to peak in the vicinity of the phase transition. We emphasize that the GAN fidelity in Eq. (14) is defined entirely in terms of the networks and can be evaluated very efficiently, once the networks have been trained.

## 3.4 Over-relaxation and models conditioned on energy

Similar to their temperature-conditioned counterparts, models conditioned on energy can also be used to provide samples directly and to study phase transitions. However, we here focus on a different application and discuss how energy-conditioned models can be integrated with MCMC to accelerate lattice simulations. Inspired by Ref. [28], where the potential of non-conditional GANs was explored as over-relaxation steps in MCMC simulations, we here propose to use *conditional* GANs for this purpose. By construction, our energy-conditioned models can provide samples with energy close to that of the current sample in the Monte-Carlo chain. As opposed to using unconditional GANs, no in general numerically expensive pre-sampling of the model is required to obtain samples within the desired energy range.

More specifically, the model we use here has the ImplicitGAN architecture introduced above. As opposed to the discussion in Sec. 3.3, where we focused on temperature-conditioned models, we here use the energy per site $e(\theta)$ of each sample $\theta$ rather than temperature as conditional input and focus on $G(z, e)$ instead of the generalized form $G(z, c)$.

### 3.4.1 General procedure

Once the models are trained we generate samples in the following way:

1. Starting from an initial configuration $\theta_0$,

2. perform $n_{\text{MC}}$ MCMC updates to obtain a configuration $\theta_t$.

3. To implement an over-relaxation step, we use the trained model and construct a new configuration, $\theta_t'$, according to $\theta_t' = G(z_t, e_t^*)$, where $e_t^*$ is obtained by fine-tuning the energy of the sample to the desired value,

$$e_t^* = \arg\min_e [E(G(z_t, e)) - E(\theta_t)]^2, \tag{15}$$

with $z_t$ being sampled from the prior distribution $P(z)$.

4. Move to step 2 until enough samples are retained.

Note that, ideally, $e = E(\boldsymbol{\theta}_t)/N^2$ would minimize Eq. (15), but this is not the case since GANs only approximately learn the distribution (see Appendix B for a discussion). Nonetheless, the energy of the samples produced by $G(z, E/N^2)$ are close to $E$ and the true optimum of Eq. (15) is expected to be in the vicinity of $E(\boldsymbol{\theta}_t)/N^2$. This makes finding $e_t^*$ more efficient in our energy-conditioned model.

While it was argued in Ref. [28] that the selection probability $W$ [entering Eq. (6)] of the GAN-based over-relaxation step is expected to be (approximately) symmetric, $W(\boldsymbol{\theta}|\boldsymbol{\theta}') = W(\boldsymbol{\theta}'|\boldsymbol{\theta})$, we emphasize that this will strictly speaking not hold in general nor exactly. For instance, GANs suffering from the mode-collapse problem will fail to lead to a symmetric $W$. Nonetheless, we here *assume* that it holds for our trained models, which allows simplifying Eq. (6) to Eq. (7) and test, in Sec. 4.5.3, whether the samples generated from it have statistical properties close to the ground truth. The validity of this assumption is supported empirically by the good performance of the models.

### 3.4.2 Solving the optimization problem

One way to solve Eq. (15) is to back-propagate the gradients through the entire generator, keeping its weights fixed, which will be very expensive as it requires multiple forward and backward passes over a deep neural network and the number of iterations may be very large. Another practical problem with this approach is that in our architecture multiple copies of conditional information are set as input to the generator. If gradient descent is used, it is possible that it may decrease some of the values and may cause others to increase. If only a single copy of conditional information is used during training, the GAN may completely ignore this conditional information among relatively larger number of noise variables.

A simpler way is to solve it as a bandit optimization problem, where the only feedback one gets is the function value $f(e) = E(G(z, e))$ and not the gradient. When the model is only conditioned on energy, the bandit version of the problem is only one dimensional. Most well-known methods existing in the literature solve this problem by constructing an unbiased estimate of the gradient of 'close approximation' of $f$ and then performing the updates from $e \to e + \Delta e$ according to gradient descent, i.e.

$$\Delta e = -\alpha(f(e_t) - E(\boldsymbol{\theta}_t))f'(e_t), \tag{16}$$

where $\alpha$ is the step size. There are several methods to obtain an estimate of the gradient for a function $f(x)$. Here we use a two-point feedback estimate [67],

$$f'(x) \approx \frac{\mathbb{E}_u[(f(x + \delta u) - f(x))u]}{\delta}. \tag{17}$$

In Eq. (17), $u \sim \mathcal{N}(0, I)$ and $\delta$ should be kept sufficiently small to obtain high accuracy, while not too small to avoid increasing the variance of the gradient estimate. Instead of computing the exact expectation value, we use a stochastic estimate with only a single realization of $u$. In this way, $E(G(z, e))$ can be made arbitrarily close to $E(\boldsymbol{\theta})$. In practice, we set a threshold value $\Delta E_{\text{thr}}$ and the optimization will be done until a configuration with $\Delta E = |E(G(z_t, e)) - E(\boldsymbol{\theta})| \leq |\Delta E_{\text{thr}}|$ is found.

When considered over multiple over-relaxation steps, the problem in Eq. (15) can also be interpreted as an online optimization problem where at time step $t$ an agent receives a loss function $f_t(e) = (E(G(e, z)) - E(\boldsymbol{\theta}_t))^2$ and the goal is to minimize the loss accumulated over various time steps. In our implementation, we exploit this nature and use the optimum of $f_t(\cdot)$ as starting value for our iterative minimization of $f_{t+1}(\cdot)$. Note that this does not induce additional correlations in our samples since $z_t$ is sampled independently at each time step.

# 4 Numerical experiments

In this section, we present a detailed study of the performance of the generative modelling approaches outlined above, using the 2D XY model as a concrete example. We first compare the model conditioned on temperature with certain baseline approaches that are defined below. In the second set of experiments, we test the ability of our model to detect phase transitions in an unsupervised way by evaluation of $\mathcal{D}(T)$ and $\mathcal{F}_{\text{GAN}}(T)$ in Eqs. (13) and (14). Then we present results for models conditioned on energy and their integration with MCMC. In the next set of experiments, we train our models only over configurations with temperatures that are below and above the critical temperature. We then test both classes of models over the complete range of temperatures, i.e., investigate how well it can interpolate over unseen temperatures near criticality.

## 4.1 Generation of training data

In this work, we use lattices of size $N \times N$, where $N = \{8, 16\}$. The training data is obtained using the MH algorithm for 32 uniformly spaced values of temperature $T$ in the range $[0.05, 2.05]$. For each value of $T$, 10000 configurations are generated. Starting from a randomly initialized state for each $T$, a sufficiently large number of configurations are rejected initially, to account for thermalization. A configuration is included in the training data set after every 120 MCMC steps for $8 \times 8$ and after 400 steps for $16 \times 16$ lattice, to reduce correlations in the training data. The angle at each lattice site is scaled down linearly from $[0, 2\pi)$ to $[0, 1)$. Thus each configuration is a 2D matrix with each entry between $[0, 1)$. The data is then characterized by investigating the distribution of observables like magnetization $\boldsymbol{m}$, energy $E$, and vorticity $V$, all as a function of $T$. The samples generated via MCMC as well as the estimated observables serve as the ground truth for evaluations.

## 4.2 Evaluation metrics

How do we know whether and to which extent the ensemble of generated configurations follow the true distribution? To evaluate, we compute the aforementioned observables using generated samples, and compare the distribution of these observables with the distribution of those obtained from MCMC simulations. To compare these distributions, we deploy the following measures on the histograms of observables generated for 500 different configurations.

### 4.2.1 Percentage overlap (%OL)

Our first measure is %OL, which corresponds to the overlap between two histograms, each of which is normalized to unit sum. Mathematically, the %OL of two distributions $P_r$ and $P_\theta$ is calculated as:

$$\%\text{OL}(P_r, P_\theta) = \sum_i \min(P_r(i), P_\theta(i)), \tag{18}$$

where $i$ is the bin index. We use 40 bins in the range $[0,1]$ for the histogram of magnetization and 80 bins in the range $[-2,0]$ for energy. It is not a self-sufficient measure in the sense that the %OL between the histograms can be quite small even though the computed values of observables are sufficiently close to each other.

### 4.2.2 Earth mover distance (EMD)

The second measure of the distance between two probability distributions we use is EMD with the following interpretation: if the distributions are thought of as two different ways of piling up a certain amount of dirt, the EMD is the minimum cost of turning one pile into the other.

Here, the cost is assumed to be the amount of dirt moved times the distance by which it is moved. The EMD $W(P_r, P_\theta)$ between two distributions $P_r$ and $P_\theta$ of a scalar observable $y$ is defined as

$$W(P_r, P_\theta) = \sum_{x=-\infty}^{\infty} \left| \sum_{y=-\infty}^{x} (P_r(y) - P_\theta(y)) \right|.$$

## 4.3 Baseline models for comparison

We perform a series of numerical experiments to test the effectiveness of the proposed methods. For comparison, we use modifications and extensions of the method of [34] as our two baselines, which provide a reference for the performance of our proposed Implicit-GAN approach.

### 4.3.1 C-HG-VAE

The first baseline model we use is C-HG-VAE. It is a prescribed generative model and was proposed in [34], referred to by them as HG-VAE. Being the (to the best of our knowledge) only available generative model which has been designed specifically for sampling the 2D XY model, it is the most natural starting point for us to construct a baseline model.

The C-HG-VAE employs CNNs instead of fully connected networks to account for translational symmetry of the physical system. To improve the agreement of thermodynamic observables with the ground truth, Ref. [34] modified the standard VAE loss function by additionally including the following term:

$$\mathcal{L}_H = [e(\boldsymbol{\theta}) - e(\hat{\boldsymbol{\theta}})]^2,\tag{19}$$

which involves the energies $e(\boldsymbol{\theta})$ and $e(\hat{\boldsymbol{\theta}})$ per lattice site of the ground truth ($\boldsymbol{\theta}$) and the generated configurations ($\hat{\boldsymbol{\theta}}$), respectively. A multivariate standard normal distribution was chosen as the prior $P(z)$ and, during training, the input spin configuration to the encoder is $\boldsymbol{s} = \{\theta_i\} \in \mathbb{R}^{N \times N}$. For the ease of implementation with standard CNN libraries, the input is formatted as two channels, one consisting of the spin configuration and the other consisting of $T$. This format has also been used by AlphaGo [68]. The output of the decoder (i.e., reconstruction layer) is split into two terms $\mu$ and $\sigma$ corresponding to the parameters of a Gaussian distribution. Configurations were generated by sampling from the Gaussian $\mathcal{N}(\mu_i, \sigma_i)$, $\mu \in \mathbb{R}^{N \times N}, \sigma \in \mathbb{R}^{N \times N}$, with each lattice site $i$ distributed independently. In the abbreviation HG-VAE, $H$ refers to the $\mathcal{L}_H$ term and $G$ to the Gaussian parametric specification of the reconstruction layer. HG-VAE generates new configurations using $z$ sampled from the approximately learned variational distribution $Q_\phi(z|x)$ and then feeds these $z$ to the decoder. Generating $z$ from $Q_\phi(z|x)$ requires use of MC samples for that corresponding temperature. Hence, their method cannot generate configurations for temperatures not in the training data. But since our goal is to generate configurations even for temperatures for which no training data is available, we modify their method to a conditional model named C-HG-VAE by providing additional information of temperature to both encoder and decoder. For generating new configurations, we provide $z \sim \mathcal{N}(0, I)$ and $T$ to the decoder. $T$ is concatenated multiple times with $z$ so as the decoder does not ignore this information along with multiple $z$. The block diagram representation of C-HG-VAE is the same as that of the C-VAE in Fig. 1a.

### 4.3.2 C-GAN

As second baseline model, we use a prescribed form of a standard C-GAN, introduced in Sec. 2.2. The C-GAN employing CNNs was trained on the space of angles to reconstruct configurations, given $T$. The input to the generator consists of $T$ concatenated with $\boldsymbol{z} \in \mathbb{R}^N$ sampled

from a Gaussian prior, where $N$ is the linear lattice size. Similar to C-HG-VAE, the generator outputs $\boldsymbol{\mu}_i \in \mathbb{R}^{N \times N}$ and $\boldsymbol{\sigma}_i \in \mathbb{R}^{N \times N}$ corresponding to the parameters of a Gaussian distribution from which the configurations are sampled. The reparametrization trick [57] is used to ensure differentiability of the network. The input of the discriminator has two channels—one consisting of the spin configurations $x$ and the other of $T$. The output of the discriminator is a scalar distinguishing the real from the fake sample.

## 4.4 Proposed method: ImplicitGAN

This is the proposed implicit C-GAN approach. While all of the key components of this method have been motivated and explained in detail in Sec. 3.2 above, we here provide a concise summary of it:

1. The angles $\theta_i$ of the spins in each sample are shifted, $\theta_i \rightarrow \theta_i + \theta_0$, such that the net magnetization vector ($\boldsymbol{m}$) always points in the direction corresponding to $\theta_i = 0$.

2. The reconstruction layer of the generator consists of two channels $[x_i, y_i]$, which we normalize at each site as $[x_i, y_i] \rightarrow [x_i, y_i]/\sqrt{x_i^2 + y_i^2}$. The input of the discriminator has 3 channels, with the first two channels consisting of cosines and sines of lattice angles and the 3rd channel containing conditional variable, $T$ or $E$.

3. To take into account the periodic boundary conditions of the lattice, we use periodic padding of size 1 for the input layer of the discriminator.

4. To minimize the biases, Eq. (9) was used as objective function. The value of $\lambda$ was chosen to be 10 for $8 \times 8$ and 1 for $16 \times 16$ lattices.

5. To maximize the entropy of the generated samples, the output layer of the discriminator now has two outputs, $A(\hat{T}|G(z, c))$ and $D(x)$, with learning objective given in Eq. (12). The value of $\gamma$ was chosen to be 100 and 10 for $16 \times 16$ and $8 \times 8$ lattices, respectively.

Below, we use "ImplicitGAN-$T$" to refer to the situation that samples are generated by the GAN trained conditioned on $c = T$. While "ImplicitGAN-$E$" indicates that sampling is performed by local-update MCMC for a given $T$ combined with over-relaxation steps with the ImplicitGAN-$E$ model as explained in Sec. 3.4.

Table 1: Evaluation metrics, as defined in Sec. 4.2, along with standard deviation, computed over 500 configurations and averaged across all temperatures. Smaller EMD and higher %OL are better. Best values are indicated in bold.

| Metric | Lattice size | C-GAN | C-HG-VAE | ImplicitGAN-$T$ |
|---|---|---|---|---|
| EMD | $8 \times 8$ | $0.358 \pm 0.246$ | $0.157 \pm 0.086$ | $\mathbf{0.038 \pm 0.024}$ |
| Magnetization | $16 \times 16$ | $0.152 \pm 0.056$ | $0.118 \pm 0.028$ | $\mathbf{0.041 \pm 0.043}$ |
| EMD | $8 \times 8$ | $0.484 \pm 0.250$ | $0.256 \pm 0.063$ | $\mathbf{0.022 \pm 0.012}$ |
| Energy | $16 \times 16$ | $0.233 \pm 0.140$ | $0.296 \pm 0.060$ | $\mathbf{0.010 \pm 0.005}$ |
| %OL | $8 \times 8$ | $29.31 \pm 33.35$ | $52.18 \pm 19.15$ | $\mathbf{76.69 \pm 6.46}$ |
| Magnetization | $16 \times 16$ | $7.97 \pm 16.39$ | $42.78 \pm 17.33$ | $\mathbf{67.34 \pm 20.41}$ |
| %OL | $8 \times 8$ | $9.43 \pm 13.94$ | $10.29 \pm 5.43$ | $\mathbf{68.28 \pm 20.72}$ |
| Energy | $16 \times 16$ | $13.64 \pm 19.33$ | $0.62 \pm 0.03$ | $\mathbf{65.83 \pm 18.35}$ |

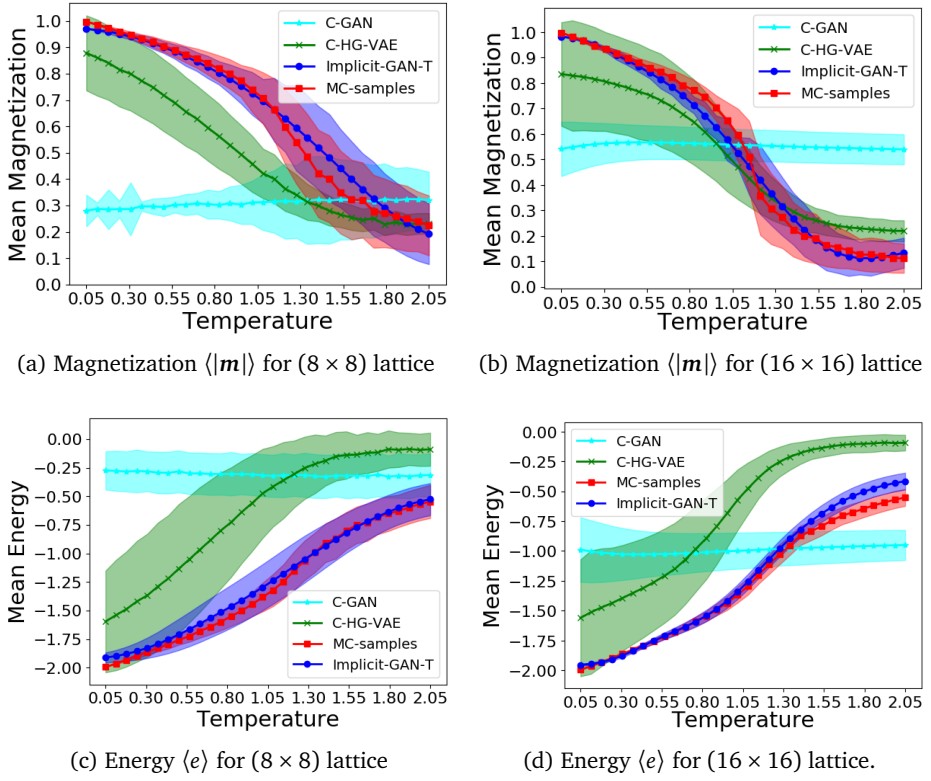

(a) Magnetization $\langle|\boldsymbol{m}|\rangle$ for $(8 \times 8)$ lattice

(b) Magnetization $\langle|\boldsymbol{m}|\rangle$ for $(16 \times 16)$ lattice

(c) Energy $\langle e \rangle$ for $(8 \times 8)$ lattice

(d) Energy $\langle e \rangle$ for $(16 \times 16)$ lattice.

Figure 2: Expectation values (dots and lines) of observables (normalized per site) computed from samples generated by the indicated methods as a function of temperature. Shaded portions represent the standard deviation of the corresponding observable. MC samples are taken as the ground truth; the method giving more overlap with the ground truth is better.

## 4.5 Results

### 4.5.1 Comparison with baselines: matching observables

For comparison with baselines, the trained temperature-conditioned models described in Sec. 4.3 were tested by computing observables, namely magnetization and energy, over the generated configurations. Fig. 2 illustrates mean magnetization $\langle|\boldsymbol{m}|\rangle$ and mean energy $\langle e \rangle$ values as a function of $T$. We can notice that $\langle|\boldsymbol{m}|\rangle$ decreases and $\langle e \rangle$ increases with $T$ for all methods except C-GAN. This shows that C-GAN fails completely to capture the statistics of the data it is supposed to generate. We can also see that the distribution of ImplicitGAN-$T$-generated observables is much closer to the ground truth (MC) as compared to that of C-HG-VAE generated observables. These results, with the metrics averaged across temperatures, are quantified in Table 1. The implicit-GAN-$T$ produces the best results over all the metrics as well as lattice sizes. Our ablation analysis presented in Appendix A shows which of the different improvements of the method were particularly crucial in enhancing the performance.

### 4.5.2 Detecting phase transitions

We now analyze the ability of the model to detect phase transitions by analyzing its susceptibility to changes in temperature using the two measures introduced in Sec. 3.3.

We begin with $\mathcal{D}$ in Eq. (13) which is plotted in Fig. 3a with $\Delta T = 0.0625$, computed over 500 configurations produced by the generator. We observe that it exhibits peaks in the

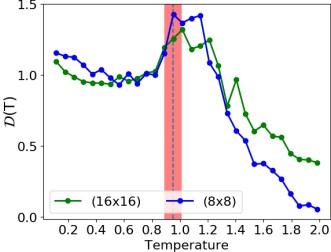
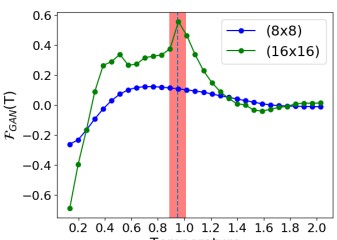
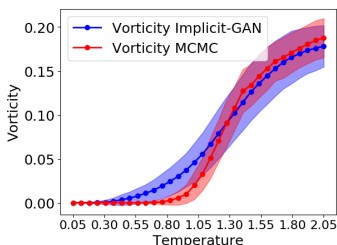

(a) $\mathcal{D}(T)$ computed across various temperatures. The peaks are observed around the critical temperature. The shaded portion is $0.950 \pm 0.0625$.

(b) $\mathcal{F}_{\text{GAN}}(T)$ computed across various temperatures.

(c) Observed vorticity for $16 \times 16$ sites as a function of temperature.

Figure 3: Detection of BKT phase transition (a,b) directly from measures defined in terms of the networks (unsupervised) and (c) by evaluation of the vorticity of generated samples.

vicinity of the expected phase transition. However, there is no clear maximum, but rather a double-peak feature. Also the finite-size scaling is opposite to what one would expect, since the double-peak features move to larger rather than smaller temperatures with increasing $N$. More dramatically, the trend does not indicate that these features approach the true location of the transition at large $N$ as they are further way from the BKT transition temperature for larger $N$.

Due to these shortcomings of $\mathcal{D}$ for detecting the BKT transition in our GAN architecture, we here focus on the second measure—the GAN-fidelity—introduced in Eq. (14) with corresponding plot in Fig. 3b, using $\Delta T = 0.0625$. For the larger system size, we here observe a clear, isolated peak very close (at around $T \approx 0.95$) to the expected transition temperature for that system size. For the smaller system size, the peak gets broader and is also shifted to the left. While the broadening is a natural feature of smaller $N$, the shift of its maximum is not the expected finite-size scaling trend—this is similar to $\mathcal{D}$, but now seems to approach the correct value with increasing $N$. One reason for the unexpected trend in the peak position could be that $\mathcal{F}_{\text{GAN}}$ is more reliable for the GAN with the larger system size: we found that, at lower $N$, the discriminator is not as successful in determining fake samples (we find $\mathbb{E}[D(G(z;T),T)]$ around 0.45 for $N = 8$ as opposed to around 0.15 for $N = 16$). Note that the negative values of $\mathcal{F}_{\text{GAN}}$ at very low $T$ are clearly unphysical and just related to the fact that the generator underestimates the magnetization slightly at low temperatures, see Fig. 2(a,b).

Notwithstanding these issues, it is encouraging to see that we can capture the phase transition without prior knowledge of the underlying relevant observable, using the simple measure $\mathcal{F}_{\text{GAN}}$ that is readily evaluated once the generative model has been trained. Further work, however, is required to see what the advantages and limitations of this approach are and to understand the finite size scaling behavior in the XY and other models. Likely, a combination with unsupervised clustering algorithms, e.g., that of [49], can provide additional assistance in detecting phase transitions in an unsupervised way. We leave a detailed, system-size-dependent study of these aspects for future work.

On top of being able to capture the phase transition in an unsupervised way, we are dealing with a generative model. Consequently, in cases where we do know the physical quantity capturing the phase transition, we can also directly compute it with the samples generate by the networks. In the case of the 2D XY model, the transition is characterized by the suppression (proliferation) of vortices when entering the low-temperature (high-temperature) phase. For

this reason, we have computed the number of vortices as a function of temperature, both in the generated and in the MCMC samples; as can be seen in Fig. 3c, we find good agreement. This shows that the Implicit-GAN approach can, indeed, capture topological excitations, which have cause problems in other applications of neural networks [44].

### 4.5.3 Models conditioned on energy

We next test the procedure introduced in Sec. 3.4 of using energy-conditioned models for over-relaxation steps in the context of the 2D XY model. In terms of training and architecture, the only difference to ImplicitGAN-$T$ is that the prior distribution was chosen to be uniform in $[-1, 1]$ (instead of Gaussian) and that $e(\boldsymbol{\theta})$ was provided as conditional information (instead of temperature), which we have shifted by 1.0 so that its mean value is around zero over the temperature range. The same training data, see Sec. 4.1, was used.

To solve Eq. (15), only 3 iterations according to Eq. (16) with $\delta = 0.075$ in Eq. (17) were performed. If the best of these 3 iterations did not yield a configuration with $\Delta E$ less than the chosen $\Delta E_{\text{thr}}$, the over-relation step was dropped. A temperature-dependent threshold $\Delta E_{\text{thr}}$ linearly increasing from $[1/N^2, 8/N^2]$ across 32 temperatures was used in our numerics. For stability purposes, gradients clipping between $[-0.02, 0.02]$ was also done.

Naturally, if only very few over-relaxation steps are performed, it will be very difficult to see in the data whether ImplicitGAN-$E$ biases the Monte-Carlo chain and leads to incorrect results. For that reason, we focused our experiments on the regime where significant biases would be apparent if they were present and performed only $n_M = 2N$ (recall $N$ is the *linear* system size) local updates in between over-relaxation steps. Nonetheless, as can be seen in Fig. 4, there is very good agreement between the ground truth (pure MCMC simulations) and our heavily GAN-over-relaxed simulations. As we show in Appendix B, the additional over-relaxation steps reduce the correlations significantly between subsequent samples in the Monte-Carlo chain.

To reduce the thermalization time, the Markov chain is intialized by generating samples from Implicit-GAN-E itself. The input $e$ needed for the initial configuration is obtained for a given $T$ by a linear approximation of the energy vs. temperature curve of MC samples (Figs. 4c, 4d). Other initializations, including random initialization, give similar results, but need higher burnout.

### 4.5.4 Interpolating across unseen temperatures around $T_c$

After having obtained architectures capable of modelling the joint distribution of spin configurations across temperatures, we next test whether these models can also generate samples in the vicinity of the phase transition without having been trained on samples in that regime—a much more challenging problem. We define the critical region as $T \in [0.75, 1.25]$. Note that the critical temperature is $T_c \approx 0.89$ [69] for large system sizes, $N \to \infty$; due to logarithmic finite-size corrections, we expect it to be larger, about 0.95, for our system sizes [44].

To test this, we trained a new ImplicitGAN model for both classes of conditional models discussed above, on the configurations for temperatures in the interval $[0.05, 0.75] \cup [1.25, 2.05]$, i.e., outside the critical region. This corresponds to a 25% reduction in training data. Then we test our model by also interpolating for the temperatures which are not even present in the training data.

The results are presented in Fig. 5, where all hyper-parameters were kept the same as before. One can see that both Implicit-GAN-$T$ and Implicit-GAN-$E$ still capture the main tendencies of the data, although the former has significantly reduced accuracy in magnetization. The performance of the latter, however, is almost unaffected. Consequently, using GANs to enhance MCMC simulations is even possible when no training data is provided in the critical region.

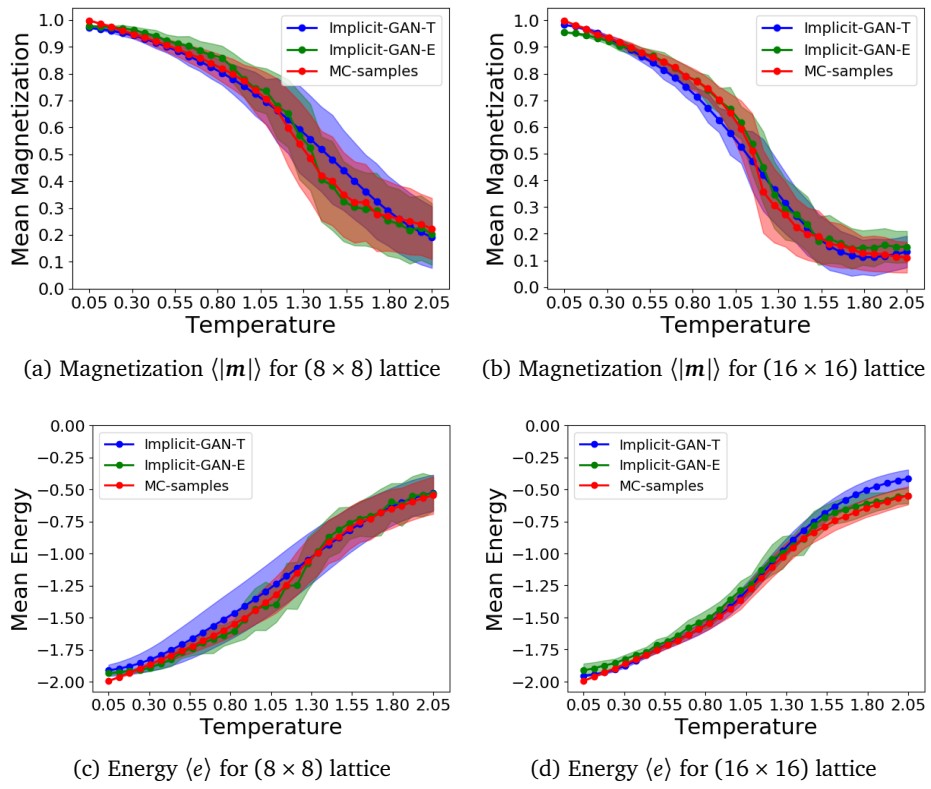

(a) Magnetization $\langle|\boldsymbol{m}|\rangle$ for $(8 \times 8)$ lattice    (b) Magnetization $\langle|\boldsymbol{m}|\rangle$ for $(16 \times 16)$ lattice

(c) Energy $\langle e \rangle$ for $(8 \times 8)$ lattice    (d) Energy $\langle e \rangle$ for $(16 \times 16)$ lattice

Figure 4: Comparison of energy-conditioned models integrated with MCMC and temperature-conditioned (direct sampling) models with ground truth (MC samples). Symbols and lines indicate average values and shaded portion the standard deviation of the corresponding observable as a function of temperature.

## 5  Conclusions

In this work, we have studied different deep-learning-based approaches for generating spin configurations. We have discussed in detail several modifications of the basic models in order to warrant a more efficient representation of the states, that, e.g., takes into account symmetries of the system and the geometry of the local degrees of freedom. Furthermore, the correlations between the samples generated by the model are shown to be reduced by incentivizing our model to increase the entropy of the learnt distribution. Although the approaches used are more generally applicable, we employed the 2D XY model to benchmark the models' performances. To this end, samples were generated using MCMC to train the models. MCMC was also used to provide the ground truth to compare the generated samples with. For the latter, we investigated the histograms of relevant observables—magnetization, energy, and vorticity. Overall, we found that implicit models perform better and, in particular, our proposed ImplicitGAN outperforms all other machine-learning models considered.

We have focused on conditional models, which, after training, can be used to generate configurations for in principle arbitrary values of tuning parameters—in our case temperature and energy. We demonstrate that this can be employed for generating configurations near criticality, even without providing training data in the vicinity of the transition. This could be useful for circumventing or at least mitigating critical slowing down in MCMC simulations. It also provides the perspective that, instead of storing a huge amount of samples for an interesting model, one could just store (and make publicly available) a precisely trained neural

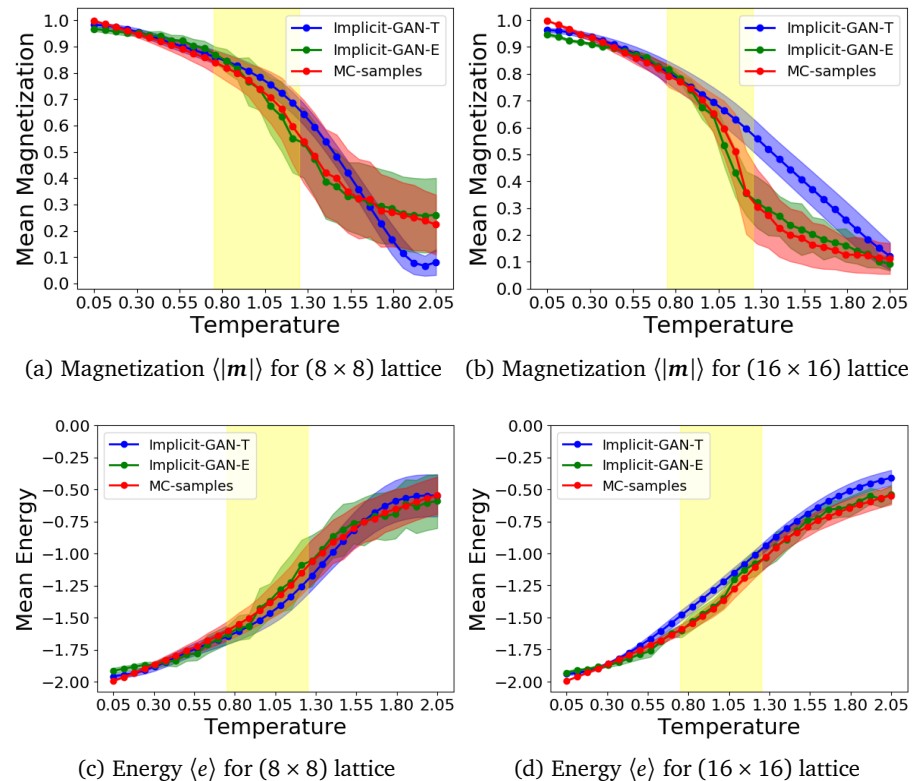

(a) Magnetization $\langle |m| \rangle$ for $(8 \times 8)$ lattice

(b) Magnetization $\langle |m| \rangle$ for $(16 \times 16)$ lattice

(c) Energy $\langle e \rangle$ for $(8 \times 8)$ lattice

(d) Energy $\langle e \rangle$ for $(16 \times 16)$ lattice

Figure 5: Same as Fig. 4, but no training data was provided in the region $T \in [0.75, 1.25]$ (highlighted in yellow) in the vicinity of the transition.

network to generate samples for future use. We further hope that, when applied to experimental data, it can be used to gain insights about parameter regimes inaccessible in the lab. For these applications, the flexibility of conditional models could prove crucial, since they allow for a multitude of possible conditional variables associated with samples, including complex post-selection criteria.

Finally, we have also shown that conditional models themselves can be employed to detect phase transitions, without any prior knowledge, by investigating the networks' susceptibility to parameter changes. Most importantly, we propose a GAN fidelity measure $\mathcal{F}_{\mathrm{GAN}}$ that can be readily evaluated for any trained GAN and is demonstrated to peak in the vicinity of transitions, in analogy to the well-known quantum fidelity measure and its thermal extensions [66]. We hope that this can supplement unsupervised clustering algorithms, such as that of Ref. [49], for future machine-learning-based studies of phase transitions. One could also explore interpretable ML models [70] to extract the crucial physical aspects, such as order parameters or defect proliferation, underlying the phase transition. On a more general level, this illustrates the advantages of the additional "tuning parameter" $c$ of conditional models, which further opens up the possibility to study phase transitions in one and the same neural network as a function of $c$. One might wonder whether (and what kind of) different universality classes of transitions can be established in conditional networks.

In the future, we are also planning to further test and refine the ImplicitGAN approach, by applying it to other classical models and systematically studying the behavior of observables and $\mathcal{F}_{\mathrm{GAN}}$ with increasing system size. Additional directions include developing models conditioned on system size and exploring quantum mechanical systems.

# Acknowledgements

**Funding information**  MS acknowledges support from the National Science Foundation under Grant No. DMR-1664842.

*Note added*—During the final stages of the completion of this project, another work appeared on arXiv [71], where a different generative ML technique is applied to the 2D XY model. The emphasis of this work is different from ours and, in particular, does not contain the analysis of implicit and prescribed models, the application as an over-relaxation step, nor that of network-based unsupervised indicators ($\mathcal{D}$ and $\mathcal{F}_{\mathrm{GAN}}$) of the phase transition, but instead relies on the helicity modulus.

# A  Ablation analysis

In this appendix, we perform a detailed ablation analysis for the temperature-conditioned models, to examine the effect of each of the components of our proposed Implicit-GAN approach, see Sec. 3 and Sec. 4.4, separately. For the sake of comparison, we average the values of the metrics defined in Sec. 4.2 across all the temperatures used in the training data and we name our models as

1. C-GAN: The standard prescribed C-GAN, which is also used as a baseline (Sec. 4.3).

2. C-GAN$_1$: A standard implicit C-GAN modeling $\theta_i$ using the angles $\theta_i$ rather than the two-component unit vectors $s_i$ as input. The generator is a deterministic function of $z$ and outputs the angles $\theta_i$.

3. C-GAN$_2$: It is same as the C-GAN$_1$ model but trained using $s_i = (\cos\theta_i, \sin\theta_i)$ as input. It also includes periodic padding of size 1 but the total magnetization of each sample of the training data was not rotated to point along the x-axis.

4. C-GAN$_3$: It is same as C-GAN$_2$ with magnetization direction normalization as in Eq. (8).

5. C-GAN$_4$: Same as C-GAN$_3$ but the training objective is now modified according to Eq. (9), in order to minimize the output bias.

6. Implicit-GAN: This is the proposed implicit C-GAN as was used in Sec. 4.4 in the main text. It is the same as C-GAN$_4$ but with the entropy-regularized objective of Sec. 3.2.2.

The performance of each of these models over the metrics is given in Table 2. A comparison between C-GAN and C-GAN$_1$ illustrates that, keeping other factors the same, implicit models perform better than prescribed models. Accounting for the continuity of the space of angles and the periodic boundary conditions further improves the performance as can be seen by comparing C-GAN$_1$ with C-GAN$_2$. Exploiting the global spin-rotation symmetry of the XY model brings further improvement in the agreement of the observables, as is visible from the performance of C-GAN$_3$. Thus, discontinuous jumps of $\theta_i$ at $2\pi$ and not taking into account periodic boundary conditions and spin-rotation symmetry seem to be important factors causing the bad performance of C-GAN$_1$ and C-GAN$_2$. Consistent with [34], we observed that this was not a serious problem when (unconditioned) GANs were trained only for a single temperature.

We see that the performance of C-GAN$_4$ is comparable to C-GAN$_3$ for the metrics in Table 2. However, one has to note that these metrics are not directly sensitive to whether the generator satisfies the constraint of total magnetization pointing along the $x$ axis, $\sum_i \sin(\theta_i)/N^2 = 0$; the additional term $\propto \lambda$ in Eq. (9) explicitly incentivizes the generator to obey the constraint. To test this, we compare the average values of the y-component of the magnetization, before

Table 2: **Ablation analysis:** Evaluation metrics, along with standard deviation, computed over 500 configurations of a $16 \times 16$ lattice, averaged across all temperatures. Smaller EMD and higher %OL are better.

| Metric | EMD Mag. | EMD Energy | %OL Mag. | %OL Energy |
|---|---|---|---|---|
| **C-GAN** | 0.304±0.113 | 0.234±0.14 | 7.969±16.394 | 16.643±13.863 |
| **C-GAN$_1$** | 0.290±0.101 | 0.212±0.122 | 20.6±21.275 | 18.381±8.303 |
| **C-GAN$_2$** | 0.136±0.04 | 0.098±0.064 | 41.181±21.295 | 35.269±23.922 |
| **C-GAN$_3$** | 0.071±0.075 | 0.034±0.028 | 67.068±16.092 | 47.25±21.815 |
| **C-GAN$_4$** | 0.043±0.038 | 0.041±0.035 | 69.275±22.586 | 37.181±22.397 |
| **ImplicitGAN-$T$** | 0.041±0.043 | 0.010±0.005 | 67.343±20.415 | 65.832±18.351 |

(C-GAN$_3$) and after (C-GAN$_4$) adding the term $\propto \lambda$. Figure 6 shows a significant reduction in the average 'bias', as with C-GAN$_4$ the curves are closer to x-axis. This can be considered as a first-order moment matching test to check whether the model learns the true distribution of the samples, which were reprocessed according to Eq. (8). The parameter $\lambda \approx 1 - 10$ was observed to work well. With a large value of $\lambda(\approx 100)$, the average bias across temperatures becomes small but the performance of the model over the metrics starts degrading. Hence, there exists a trade-off between the performance and bias.

Finally, we can see in Table 2 that the performance of Implicit C-GAN, in terms of reproducing the distribution of observables, is comparable to that of C-GAN$_3$ and C-GAN$_4$ for magnetization and seems to become even better for the energy. On top of that, the key advantage of the Implicit-GAN is that it generates more uncorrelated samples as compared to the latter. To quantify this, we measure correlations between a pair of samples, $\boldsymbol{\theta} = \{\theta_j\}$ and $\boldsymbol{\theta}' = \{\theta_j'\}$, generated by our models. To this end, we introduce

$$\kappa(T) = \frac{1}{N^2} \sum_j \left| \mathbb{E}\left[ e^{i(\theta_j - \theta_0)} e^{-i(\theta_j' - \theta_0')} \right] \right| \tag{A.1}$$

as our measure for the *average cross-correlation*. Here, $\theta_0 = \sum_j (\theta_j/N^2)_{2\pi}$ and $\theta_0' = \sum_j (\theta_j'/N^2)_{2\pi}$ to make sure that we do not get $\kappa \approx 0$ simply because we have exploited the global spin-rotation symmetry, see Sec. 3.1.1. The expectation value in Eq. (A.1) is taken with respect to the configurations generated by the models.

For instance, from C-GAN$_3$ to Implicit-GAN, we obtain an improvement from $\kappa = 0.65 \pm 0.38$ to $\kappa = 0.27 \pm 0.2$ at $T = 1.5$ and for $N = 16$. We observed a significant reduction in cross-correlation as compared to C-GAN$_3$ for both $8 \times 8$ and $16 \times 16$ lattices and across temperatures. Nonetheless, a comparison with the ground truth (MC) still reveals an enhanced $\kappa$ in the disordered high-temperature phase, which means that the Implicit-GAN generated samples are not perfect and do not completely explore the state space.

# B  Characteristics of ImplicitGAN-$E$

We here present more details on the properties of our ImplicitGAN-$E$ models. As already mentioned in Sec. 3.4, GANs learn distributions only approximately. As such, the energy of the states generated by $G(z, e)$, $z \sim P(z)$, will have energy densities only approximately equal to $e$. To quantify this, we plot in Fig. 7a the I/O characteristic of our ImplicitGAN-$E$ models, i.e., the distribution of $E(G(z, e))/N^2$, $z \sim P(z)$, as a function of $e$. As can be seen, $E(G(z, e))/N^2$ and $e$ clearly follow each other, but systematic deviations exist. These observations show that the use of conditional models can accelerate the search for states with the desired energy

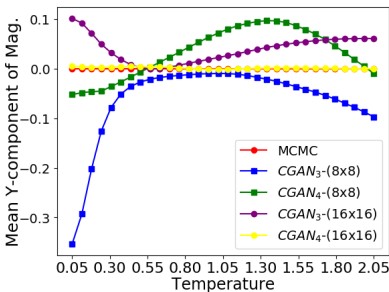

Figure 6: Average value of Y-component of magnetization computed over 500 configurations. Due to pre-processing of the MCMC data, the curves should be close zero.

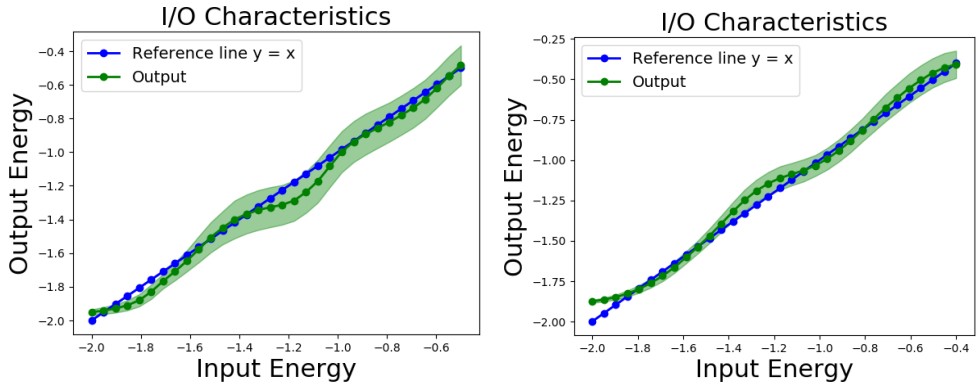

(a) Input-Output Characteristics for $(8 \times 8)$ and $(16 \times 16)$ lattices respectively.

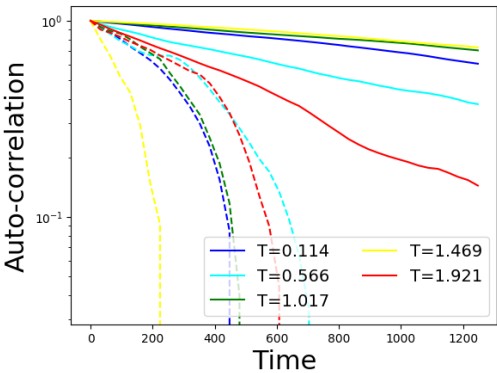

(b) Auto-correlation, as defined in Eq. (B.1), as a function of the number of local updates for 5 different temperatures and $N = 16$. Solid lines are MCMC with local updates and dashed lines is MCMC with over-relaxation.

Figure 7: Characteristics of our ImplicitGAN-$E$ model.

significantly, as compared to regular GANs. At the same time, we also learn that fine-tuning $e$ to obtain the required energy with high precision via Eq. (15) is still important; otherwise, we would obtain systematic deviations in our Markov chain.

To explicitly demonstrate that the use of ImplicitGAN-$E$ as over-relaxation steps decreases the correlations between samples in the Markov chain, we here compute the following auto-correlation function:

$$R_m(\tau) = \left[ \frac{\sum_{i=1}^{M-\tau} m_i m_{i+\tau} - (M-\tau)\langle m \rangle_{[1,M-\tau]} \langle m \rangle_{[\tau+1,M]}}{\langle m^2 \rangle_{[1,M]} - \langle m \rangle_{[1,M]}^2} \right],$$ (B.1)

where $m_i$ denotes the value of $|\boldsymbol{m}|$ in the $i$th sample in the Markov chain and $\langle m \rangle_{[j_1, j_2]} = \frac{1}{|j_2 - j_1 + 1|} \sum_{j=j_1}^{j_2} m_j$. A plot of the auto-correlation function $R_m(\tau)$ with and without the over-relaxation step is shown in Fig. 7b. Clearly, at all temperatures, the addition of GAN-based over-relaxations steps significantly reduces the correlations of samples in the Markov chain.

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
