# Peer review of "Conditional generative models for sampling and phase transition indication in spin systems"

_SciPost Physics, doi:SciPost Phys. 11, 043 (2021)_

## Round 1 · Referee Report · Anonymous (Referee 1) · 2021-5-11

Strengths

  1. In the manuscript, authors first introduced the Generative Adversarial Networks (GANs) as a general tool to explore the topological phase transition emerges in continuous many-body systems.

  2. The conditional GANs(c-GANS) show a good performance in generating samples unsupervisedly, which could reduce the auto-correlations on the Markov-Chain efficiently.

  3. Two measures were proposed and first validated in detecting the BKT phase transition unsupervisedly.

Weaknesses

  1. The classical 2-D XY model shows the magnetization in the thermodynamic limit is zero, thus, it is not so convincing to show magnetization changes with the Temperature only for lattice size (16$\times$16 and 8$\times$8 ) in fig 2.

  2. In ref.44, the authors proposed a sawtooth-type CNN filter to recognize the vortices hidden in configurations, but it still needs pre-training. One of my concerns is that, could we find a similar inner structure in GANs automatically? If not, besides the two measures inspired by thermodynamics, where is the feasibility of the GANs from?

  3. From fig.2 to fig.5, the comparison between MCMC-samples and GANs (containing I-GAN-T and cGANs) reveals distinct mismatches. It needs more explanations from technical view or better from physical view.

Report

In general, it is an exciting work which introduces the GANs into detecting topological phase transitions in many-body systems. The authors carefully investigated the applications of the GANs in both the efficiency of numerical computations and unsupervised detection of phase transitions.

Except the ref.70, at this time there is no similar work. And as emphasized by the authors in their manuscript, they adopted different technical and analytical frameworks compared with ref.70.

I recommend accepting this manuscript as an article in the journal.
  • validity: high
  • significance: high
  • originality: top
  • clarity: good
  • formatting: excellent
  • grammar: excellent

Author:  Mathias Scheurer  on 2021-07-28  [id 1621]

(in reply to Report 1 on 2021-05-11)

We are grateful for the positive evaluation of our manuscript, for judging it to be "an exciting work'', and for recommending publication in SciPost. We next address the three questions (weaknesses) mentioned by the referee.

1–Thanks for pointing this out. This work is our first step towards showing the efficacy of conditional generative adversarial networks (GANs) for modeling many-body systems. As shown in the experiments included in the manuscript, standard machine learning techniques (see baseline models defined in Sec. 4.3) do not work well for the XY model, even for small system sizes. This is why our primary focus in this work is to improve the performance of generative modelling techniques to be able to capture crucial features, such as vortex excitations, of the 2D XY model. Being a first step in an anticipated series of future works (by us and hopefully others) in this direction, we chose to focus on smaller system sizes.
Our current observations hint at the ability of the proposed GANs to model the distribution of configurations of larger system sizes well enough so as to allow for accurate prediction of magnetization as temperature and/or energy-conditioned models. In fact, one can see by comparison of Fig. 2(a) and (b) that the temperature $T_{1/2}(L)$ at which the magnetization reaches half its maximum value decreases with system size $L$.
We are planning to develop these GANs further in the future so as to efficiently handle larger lattices. The explicit demonstration that this will allow us to accurately capture the expected vanishing of $T_{1/2}(L)$ with $L\rightarrow \infty$ will be left for future work.
In this context, we further mention that, while the present work conditions the GANs on temperature and energy, we would also like to develop ways to condition the GANs on the size of the lattice.
As we completely agree with the referee that enlarging the system size is an important aspect that needs to be addressed in future research, we mention this explicitly in the conclusion (Sec. 5) by revising the last paragraph as follows:
"In the future, we are also planning to further test and refine the ImplicitGAN approach, by applying it to other classical models and systematically studying the behavior of observables and $\mathcal{F}_{\text{GAN}}$ with increasing system size. Additional directions include developing models conditioned on system size and exploring quantum mechanical systems."

2–Whether we see such inner structure in GANs is difficult to say. Interpretable ML [Mudoch et al., Ref. 70 in the revised manuscript] is an active area of current research, which could be explored in future work. At least, upon inspection of the network parameters of our trained generator and discriminator, we could not immediately develop an intuitive picture for how our GAN architecture is able generate samples; this might be particularly challenging for our models, as we use conditional GANs, which are able to generate samples not only at a single temperature (or energy) but as a function of temperature (energy).
We emphasize that, while Ref. 44 also deals with the 2D XY model, the actual machine-learning tasks and training procedures are quite different in the two works: the problem addressed in Ref. 44 is a semi-supervised way of detecting phase transitions from samples; their machine-learning setup is not built to (and hence incapable of) generating samples. Our manuscript, on the other hand, primarily addresses the problem of generating samples for a given temperature (or energy). The ML model we use are therefore generative ones (GANs) instead of classification models. We then show that GANs, once able to generate reliable samples, are well suited to also capture phase transitions in an unsupervised way. The training procedure is, however, very different and, as such, it is hard to compare the two works.
We thank the referee for bringing out this important point. Since we think that it is important, we mention it in the conclusion section of the revised version of the manuscript:
"One could also explore interpretable ML models [Mudoch et al.] to extract the crucial physical aspects, such as order parameters or defect proliferation, underlying the phase transition.''
Additionally, we believe that conditional GANs are a better choice because they need to be trained only once while the sawtooth-type CNN filter needs to be trained many times for different values of temperature. Although our GAN architecture has more parameters (approximately two times: for Generator and Discriminator) than a standard neural network as in Ref. 44, we believe that training a conditional GAN is computationally more effective for our task in the manuscript than a sawtooth-type architecture like in Ref. 44.

3–The "C-GAN'' used as baseline uses the angles $\theta_i$ as input. The same representation of the spin configurations was also used by Ref. 34. While the proposed model, "ImplicitGAN'', uses $(\sin(\theta_i), \cos(\theta_i))$ and also exploits the symmetry under global rotation of all spins (Eq. 8)---both seem to be important factors causing the bad performance of C-GANs. This is evident in the Ablation analysis (see Appendix A) where we observe incremental improvements in C-GAN as we incrementally add these modifications. Distinct mismatches are also visible in ImplicitGAN-T. As explained in manuscript, it is because of the fact that GANs only approximately learn the distribution and hence there exist some biases and small mismatches for some temperatures; especially near phase transition the distribution is quite sensitive to input. This was the main motivation behind the proposed Implicit-GAN-E which is more robust to such mismatches.

To explain it better in the manuscript, we added the following in the revised manuscript's Appendix A: Ablation Analysis:
"Thus, discontinuous jumps of $\theta_i$ at $2\pi$ and not taking into account periodic boundary conditions and spin-rotation symmetry seem to be important factors causing the bad performance of C-GAN$_1$ and C-GAN$_2$. Consistent with [34], we observed that this was not a serious problem when (unconditioned) GANs were trained only for a single temperature.''

---

## Round 1 · Referee Report · Anonymous (Referee 2) · 2021-5-15

Strengths

  1. In this work generative adversarial networks (GANs) are used in a novel way to learn the distribution of spin configurations and to generate samples, in an unsupervised form.

  2. An interesting aspect of the work is the design of architectures for the generation of samples conditioned by external parameters. This avoids separate (and expensive) training for each value of the external parameters.

  3. The proposed sampling methods via implicit GANs are applied to the ferromagnetic XY model on the square lattice, considering temperature and energy as external parameters (separately). This is a paradigmatic case, where topological excitations, BKT transition, and absence of long-range order are exhibited. However, the proposed technique offers the potential to be applied to other models, both at a classical and quantum level.

  4. The work details and justifies the type of neural network used, the context, and previous research. In addition, a summary of the main properties of the generative networks used and the XY model is carried out.

  5. To evaluate the performance of the proposed models, a detailed comparative study is carried out. The magnetization and energy versus temperature curves are computed on finite-size samples obtained by means of the proposed implicit GANs, and other generative models based on variational autoencoders (VAEs), along with Monte Carlo simulations (considered the ground truth). For this, two statistical estimators of similarity of distributions are used: percentage overlap (% OL) and earth mover distance (EMD).

Weaknesses

  1. Panels (a) and (b) of figures 2 and 4 show the magnetization curve calculated by different methods. In these figures it is observed that the implicit GAN-T (fig. 2) and implicit GAN-E (fig. 4) obtain the best performance for sizes 8x8 (panels a) and 16x16 (panels b). The presence of magnetization is a finite size effect, which tends to disappear as the lattice size increases. It would be interesting to explore how the performance of implicit GANS depends on size, both in terms of these observables (magnetization and energy) and vorticity.

  2. The work highlights the usefulness of conditional GANS to detect phase transitions. For this, measures of the susceptibility of the network to changes in the external parameters are used. These indicators show peaks near the transition. Figure 3 (panels a and b) shows the evaluation of these indicators for two different sizes. The dependence of the peak position with lattice size in these figures is not entirely clear. Besides, for the first estimator D (defined in eq. 13) a double peak structure is observed. Since the scaling of the transition temperature Tc with the size is well established (ref. 42) it would be particularly useful to explore how the indicators used here (in particular GAN fidelity) predict the known scaling properties. For that, going to larger sizes, such as 32x32 or 64X64 could be clarifying. While this is probably beyond the scope of the paper, it would give the tool much more robustness.

Report

The authors present an interesting work, where they explore the possibilities of generative models in a novel way (with some parallels with the work of ref. 70).

The use of conditional GANs in this paper is oriented both towards the unsupervised generation of samples (implicit GAN-T), as well as accelerators of lattice simulations when combined with standard MCMC methods (implicit GAN-E).

The proposed unsupervised indicators to detect phase transitions in classical and quantum models are of potential interest.

Finally, the possibility of generating samples beyond the training data is relevant to explore situations outside the scope of the available experiments or in regions of parameter space where simulations become more expensive, such as in the vicinity of a phase transition.

For all the above, I recommend the publication of this work as an article in SciPost.
  • validity: good
  • significance: high
  • originality: high
  • clarity: high
  • formatting: good
  • grammar: excellent

Author:  Mathias Scheurer  on 2021-07-28  [id 1620]

(in reply to Report 2 on 2021-05-15)

We are grateful for the positive evaluation of our work, for describing it as "interesting'', and for recommending its publication in SciPost. We next address the two questions (weaknesses) mentioned by the referee.

1–As this aspect is closely related to the first weakness mentioned by the first referee, we provide a very similar answer. We view this work as our first step towards showing the efficacy of conditional generative adversarial networks (GANs) for modeling many-body systems. As shown in the experiments included in the manuscript, standard machine learning techniques (see baseline models defined in Sec. 4.3) do not work well for the XY model, even for small system sizes. This is why our primary focus in this work is to improve the performance of generative modelling techniques to be able to capture crucial features, such as vortex excitations, of the 2D XY model. Being a first step in an anticipated series of future works (by us and hopefully others) in this direction, we chose to focus on smaller system sizes.
Our current observations, however, hint at the ability of the proposed GANs to model the distribution of configurations of larger system sizes well enough so as to allow for accurate prediction of magnetization, energy, and vorticity---as temperature and/or energy-conditioned models. As alluded to by the referee, one can see by comparison of Fig. 2(a) and (b) that the temperature $T_{1/2}(L)$ at which the magnetization reaches half its maximum value decreases with system size $L$.
We are planning to develop these GANs further so as to efficiently handle larger lattices and study the $L$ dependence of magnetization, energy, and vorticity systematically.
In this context, we further mention that, while the present work conditions the GANs on temperature and energy, in the future, we would also like to develop ways to condition the GANs on the size of the lattice.
As we completely agree with the referee that enlarging the system size is an important aspect that needs to be addressed in future research, we mention this explicitly in the conclusion (Sec. 5) by revising the last paragraph as follows:
"In the future, we are also planning to further test and refine the ImplicitGAN approach, by applying it to other classical models and systematically studying the behavior of observables and $\mathcal{F}_{\text{GAN}}$ with increasing system size. Additional directions include developing models conditioned on system size and exploring quantum mechanical systems."

2–First, we like to clarify our motivation for studying two measures, $\mathcal{D}$ and $\mathcal{F}_{\text{GAN}}$, for detecting phase transitions from network parameters: $\mathcal{D}$ is basically the one of Refs. 53,56; in our GAN setup, as pointed out by the referee, it does not work well as it exhibits double-peak features. Therefore and since it is qualitatively related to the widely employed quantum fidelity, we proposed (to the best of our knowledge for the first time) another measure, $\mathcal{F}_{\text{GAN}}$, which is unique to and very quickly evaluated by GANs. As can be seen in Fig. 3, it does indeed work better than $\mathcal{D}$. As this is an important point, we have improved the discussion of these measures in Sec. 4.5.2 in the revised manuscript.
Second, we agree with the referee that a systematic analysis of both $\mathcal{F}_{\text{GAN}}$ and $\mathcal{D}$ as a function of system size $L$ and the associated scaling of the peak position with $L$ are worthwhile; however, as alluded to in our response to point 1. above and as mentioned by the referee, this is beyond the scope of the paper and left for future work.

---

## Round 2 · Referee Report · Anonymous · 2021-8-9

Report

Dear Editor
I have reviewed the changes made to the manuscript. I consider that in its current form the paper should be accepted for publication.

---

## Round 2 · Referee Report · Anonymous · 2021-8-11

Report

Dear Editor,
My comments and suggestions have been carefully considered in the latest manuscript.
Meanwhile, authors’ responses are accurate and clear to the questions.
Thus, I recommend to accept the paper as an formal article on SciPost.

---

## Round 2 · List of Changes

Change #1. Section 5. Conclusion. Paragraph 3.
One could also explore interpretable ML models [Murdoch et al.] to extract the crucial physical aspects, such as order parameters or defect proliferation, underlying the phase transition.

Change #2. Section 5. Conclusion. Paragraph 4.
In the future, we are also planning to further test and refine the ImplicitGAN approach, by applying it to other classical models and systematically studying the behavior of observables and $\mathcal{F}_{\text{GAN}}$ with increasing system size. Additional directions include developing models conditioned on system size and exploring quantum mechanical systems.

Change #3. Appendix A. Ablation analysis. Paragraph 2.
Thus, discontinuous jumps of $\theta_i$ at $2\pi$ and not taking into account periodic boundary conditions and spin-rotation symmetry seem to be important factors causing the bad performance of C-GAN$_1$ and C-GAN$_2$. Consistent with [34], we observed that this was not a serious problem when (unconditioned) GANs were trained only for a single temperature.

Change #4. Section 4.5.2 Detecting phase transitions. Paragraph 2
The line "A more detailed finite-size scaling analysis would be required to address this issue" is removed and the line "We leave a detailed, system-size-dependent study of these aspects for future work" is added later in this section.

The first line of third paragraph of this section now reads as "Due to these shortcomings of $\mathcal{D}$ for detecting the BKT transition in our GAN architecture, we here focus on the second measure---the GAN-fidelity---introduced in Eq. (14) with corresponding plot in Fig. 3b, using $\Delta T = 0.0625$.

(Minor) Change #5. Section 5. Conclusion. Paragraph 2.
The line ``We demonstrate that this can be used to generate configurations near criticality..."" is modified as ``We demonstrate that this can be employed for generating configurations near criticality...''

(Minor) Change #6. Section 5. Conclusion. Paragraph 3.
The line ``Most importantly, we propose a GAN fidelity measure that can be readily ...'' is modified as
``Most importantly, we propose a GAN fidelity measure $\mathcal{F}_{\text{GAN}}$ that can be readily ...''

---

## Editorial Decision

published